# The molecular coupling between substrate recognition and ATP turnover in a AAA+ hexameric helicase loader

Neha Puri[1†‡], Amy J Fernandez[1†], Valerie L O'Shea Murray[1,2], Sarah McMillan[3], James L Keck[3], James M Berger[1*]

[1]Department of Biophysics and Biophysical Chemistry, Johns Hopkins School of Medicine, Baltimore, United States; [2]Saul Ewing Arnstein & Lehr, LLP, Centre Square West, Philadelphia, United States; [3]Department of Biomolecular Chemistry, University of Wisconsin School of Medicine and Public Health, Madison, United States

**Abstract** In many bacteria and eukaryotes, replication fork establishment requires the controlled loading of hexameric, ring-shaped helicases around DNA by AAA+(ATPases Associated with various cellular Activities) ATPases. How loading factors use ATP to control helicase deposition is poorly understood. Here, we dissect how specific ATPase elements of *Escherichia coli* DnaC, an archetypal loader for the bacterial DnaB helicase, play distinct roles in helicase loading and the activation of DNA unwinding. We have identified a new element, the arginine-coupler, which regulates the switch-like behavior of DnaC to prevent futile ATPase cycling and maintains loader responsiveness to replication restart systems. Our data help explain how the ATPase cycle of a AAA+-family helicase loader is channeled into productive action on its target; comparative studies indicate that elements analogous to the Arg-coupler are present in related, switch-like AAA+ proteins that control replicative helicase loading in eukaryotes, as well as in polymerase clamp loading and certain classes of DNA transposases.

*For correspondence:
jberge29@jhmi.edu

[†]These authors contributed equally to this work

Present address: [‡]FogPharma, Cambridge, United States

## Introduction

The proliferation of all cells relies on the accurate and timely copying of DNA by large macromolecular assemblies termed replisomes. Cellular replisomes are complex, multi-functional assemblies that co-localize a number of essential biochemical functions to increase replication efficiency and speed (*O'Donnell et al., 2013*). One such function is the unwinding of parental chromosomes to expose template DNA for leading and lagging strand synthesis, a process carried out by specialized, ring-shaped enzymes known as helicases. In Gram-negative bacteria, an enzyme known as DnaB (named DnaC in Gram-positive organisms) serves as the principal helicase for driving replication fork progression (reviewed in *Chodavarapu and Kaguni, 2016*). DnaB proteins form hexameric rings and are generally incapable of spontaneously associating with chromosomal replication start sites (origins). As a consequence, DnaB helicases are specifically recruited and deposited onto single-stranded origin DNA by dedicated initiation and loading factors.

Replicative helicase loading in bacteria depends on at least one of two known types of proteins. One widespread class, comprising factors related to a protein known as DciA, is ATP-independent (*Brézellec et al., 2016*; *Mann et al., 2017*). The second class is composed of DnaC/DnaI-family proteins, which are ATP-dependent and present in model organisms such as *E. coli* and *Bacillus subtilis* (*B. subtilis*) (*Brézellec et al., 2016*). Both DnaC and DnaI consist of an N-terminal helicase-binding domain (HBD) fused to a C-terminal AAA+ (ATPases Associated with various cellular Activities) nucleotide-binding fold (*Figure 1A*; *Ludlam et al., 2001*; *Neuwald et al., 1999b*). The AAA+ region

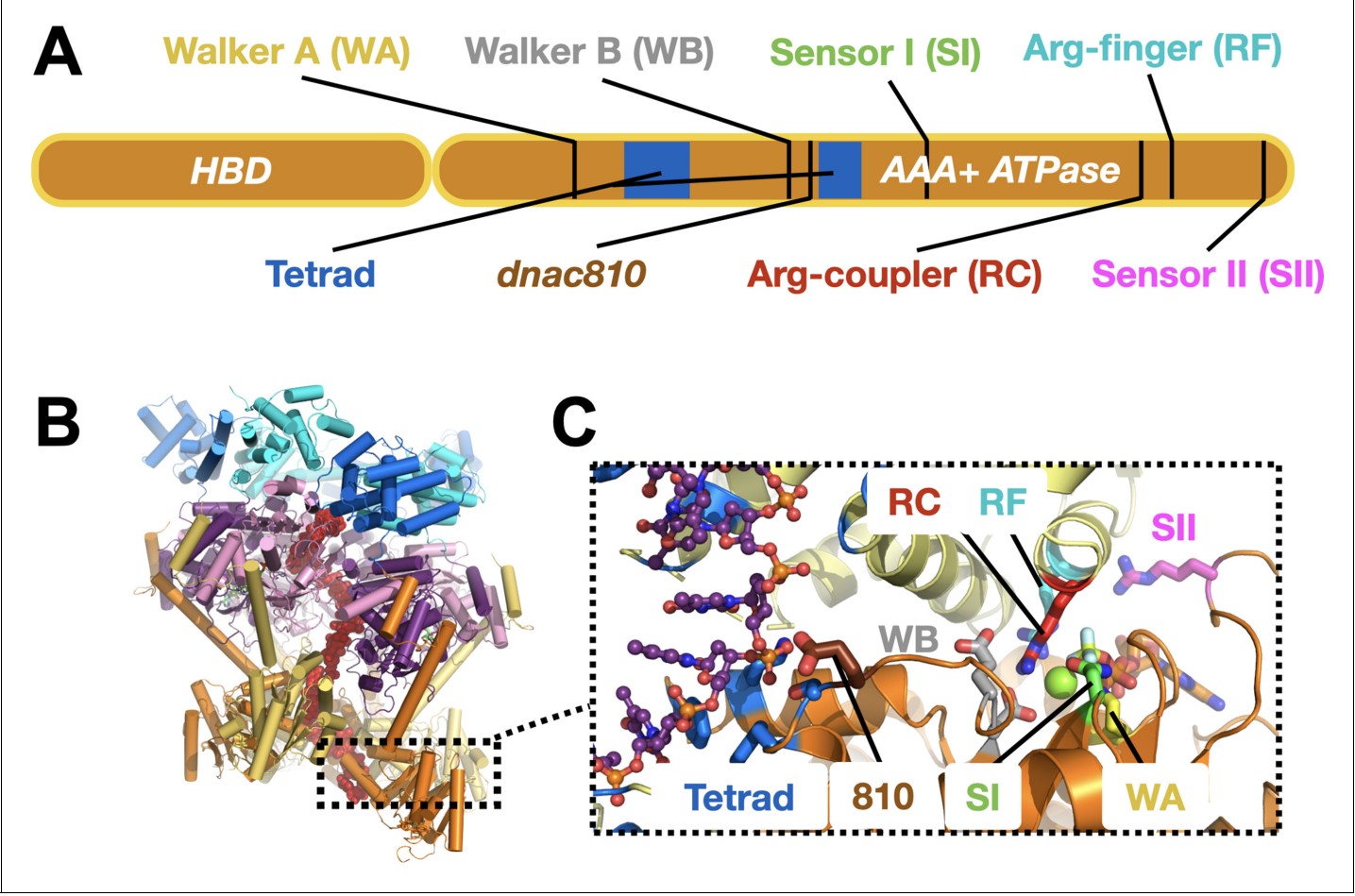

**Figure 1.** DnaC functional elements. (**A**) Domain organization of DnaC. The helicase-binding domain (HBD) spans residues 1–75 of the full-length protein. (**B**) Structure of a single-stranded DNA-bound DnaBC complex from *E. coli* shown as a cartoon view (**Arias-Palomo et al., 2019**). DnaB is colored blue (NTD) and purple (RecA ATPase domain), DnaC subunits are alternatingly orange and yellow, and ssDNA is red. (**C**) Close-up cartoon view of the DnaC active site showing elements mutated in this study. One protomer is colored orange, the other yellow; DNA is red. Motifs: WA – Walker A (Lys112); WB – Walker B (E170); SI – Sensor I (Asn203); RC – Arginine Coupler (Arg216); RF – Arginine Finger (Arg220); SII – Sensor II; 810 – Glu176; Tetrad (ssDNA-binding locus) – Lys143, Phe146, Ser177, Tyr179.

The online version of this article includes the following video for figure 1:

**Figure 1—video 1.** Morph movie going from Apo (DNA-free) to DNA-bound states of DnaB-bound DnaC, showing a close-up of the DnaC ATPase site and the movement of the Arg-coupler.

https://elifesciences.org/articles/64232#fig1video1

of DnaC contains numerous signature sequence motifs (Walker A, Walker B, Sensor I and II, Arginine-finger), several of which have been reported to be critical for the function of the loader in *E. coli* (**Makowska-Grzyska and Kaguni, 2010**; **Mott et al., 2008**). A few of these elements have been examined and found to be important for controlling aspects of DnaC activity in vitro as well (**Davey et al., 2002**; **Makowska-Grzyska and Kaguni, 2010**; **Mott et al., 2008**); however, a systematic analysis of their individual mechanistic impact on the efficiency of DNA loading and unwinding by DnaB has been lacking.

DnaC-type loaders help place DnaB hexamers on single-stranded DNA by opening the helicase ring and allowing a nucleic acid strand to laterally enter into the pore of the complex (**Arias-Palomo et al., 2019**). In the context of DnaB, six DnaC protomers form a cracked hexameric oligomer, generating composite ATPase active sites at five of the six subunit interfaces (**Figure 1B, C**; **Arias-Palomo et al., 2013**; **Arias-Palomo et al., 2019**). The AAA+ fold of DnaC binds ssDNA directly, at a site distal from its ATP-binding locus (**Arias-Palomo and Berger, 2015**; **Ioannou et al.,**

*2006*; *Mott et al., 2008*). Prior to binding DNA, DnaC protomers latch onto a DnaB hexamer through their HBDs to promote helicase ring opening. In this configuration, electron microscopy studies have shown that DnaB resides in an ADP-like state and DnaC in an ATP-bound form (*Arias-Palomo et al., 2013*). DNA binding induces a large conformational rearrangement in the DnaB hexamer and compresses the DnaC ATPase ring, switching the ATPase status of the helicase and the loader into ATP and ADP forms, respectively. Curiously, although DnaB helps promote the local co-association of neighboring DnaC AAA+ domains, the ATPase activity of the loader does not manifest until ssDNA is bound (*Arias-Palomo et al., 2013*; *Davey et al., 2002*; *Kobori and Kornberg, 1982*). How DnaB promotes DnaC self-assembly yet does not directly trigger ATPase turnover by the loader is not understood. How DNA binding is relayed to the DnaC ATPase site and whether DnaB ATPase function is important for loading (as structural work has suggested) are similarly unknown. Addressing these questions is important for understanding how replicative helicase loaders exploit nucleotide-coupling mechanisms to generate productive action on client substrates, a question relevant to AAA+ ATPases in general.

Here, we have shown through a structure-guided biochemical approach that defects in different DnaC ATPase elements have distinct impacts on the ability of the loader to deposit DnaB onto DNA and to activate helicase unwinding of forked substrates. DnaC is shown to be capable of promoting helicase unloading in an ATP-dependent manner; both loading and unloading are additionally found to be influenced by the ATPase activity of the helicase as well. We have identified a set of residues in DnaC that couple loader ATPase activity to the binding of DnaB and ssDNA and that unexpectedly modulate the responsiveness of DnaC to a replication restart factor, and we have shown that analogs of one of these elements, an active site arginine, are present in related AAA+ superfamily members that operate with switch-like properties. Our data collectively explain how the ATPase status of DnaC is coupled to the recognition of its DnaB and ssDNA substrates to ensure tight coupling of helicase loading and DNA unwinding with nucleotide turnover.

## Results

### Conserved AAA+ motifs support DnaC ATPase activity

We first set out to address the extent to which five canonical AAA+ motifs support the ATP-hydrolysis activity of *E. coli* DnaC. Using site-directed mutagenesis, we generated a panel of individual DnaC variants lacking one of the five key residues found in AAA+ proteins (Walker A (WA) – K112R; Walker B (WB) – E170Q; Sensor I (SI) – N203A; Arg-finger (RF) – R220D; Sensor II (SII) – R237D). We expressed and purified each of these mutants and, using a coupled assay, tested them for ATP-hydrolysis activity in the presence of a hydrolysis-defective DnaB mutant (DnaB$^{KAEA}$) and ssDNA. The ablation of any of the five residues was found to result in the complete loss of ATPase function (*Figure 2A*). These results highlight the importance of AAA+ signature residues to the nucleotide turnover capabilities of *E. coli* DnaC, and further establish that the hydrolysis activity present in the assay derives from the loader and not the helicase or a contaminating ATPase.

### DnaC ATPase function is coupled to helicase loading efficiency

Since the ATPase activity of DnaC requires the presence of DnaB (*Arias-Palomo and Berger, 2015*; *Davey et al., 2002*), we next asked how the ATPase cycle of the loader contributes to productive interactions with its client helicase. The role of each DnaC AAA+ ATPase active site residue was investigated in the loading of DnaB onto a closed-circular M13mp18 ssDNA substrate using a gel-filtration chromatography assay that resolves ssDNA-DnaB-DnaC complexes, free DnaB, and DnaC (*Davey et al., 2002*; *Figure 2B*). As controls, efficient DnaB loading was observed in the presence of wild-type DnaC (*Figure 2C*), whereas little ssDNA association was detected for DnaB in the absence of DnaC (*Figure 2—figure supplement 1A*). For comparison, the loading efficiency of the DnaC HBD, which wholly lacks any ATPase function and has been shown previously to catalyze DnaB deposition onto ssDNA (*Arias-Palomo et al., 2013*), was approximately half that of WT DnaC (*Figure 2D*, *Figure 2—figure supplement 1B*). Investigation of the remaining active site mutants shows that they are largely intermediate in activity between the HBD and WT, with the activity of the Sensor I mutant proving the most compromised (slightly more than the HBD alone) (*Figure 2D*, *Figure 2—figure supplement 1C*). The loading activities of the Walker-B and Arg-finger mutants were roughly

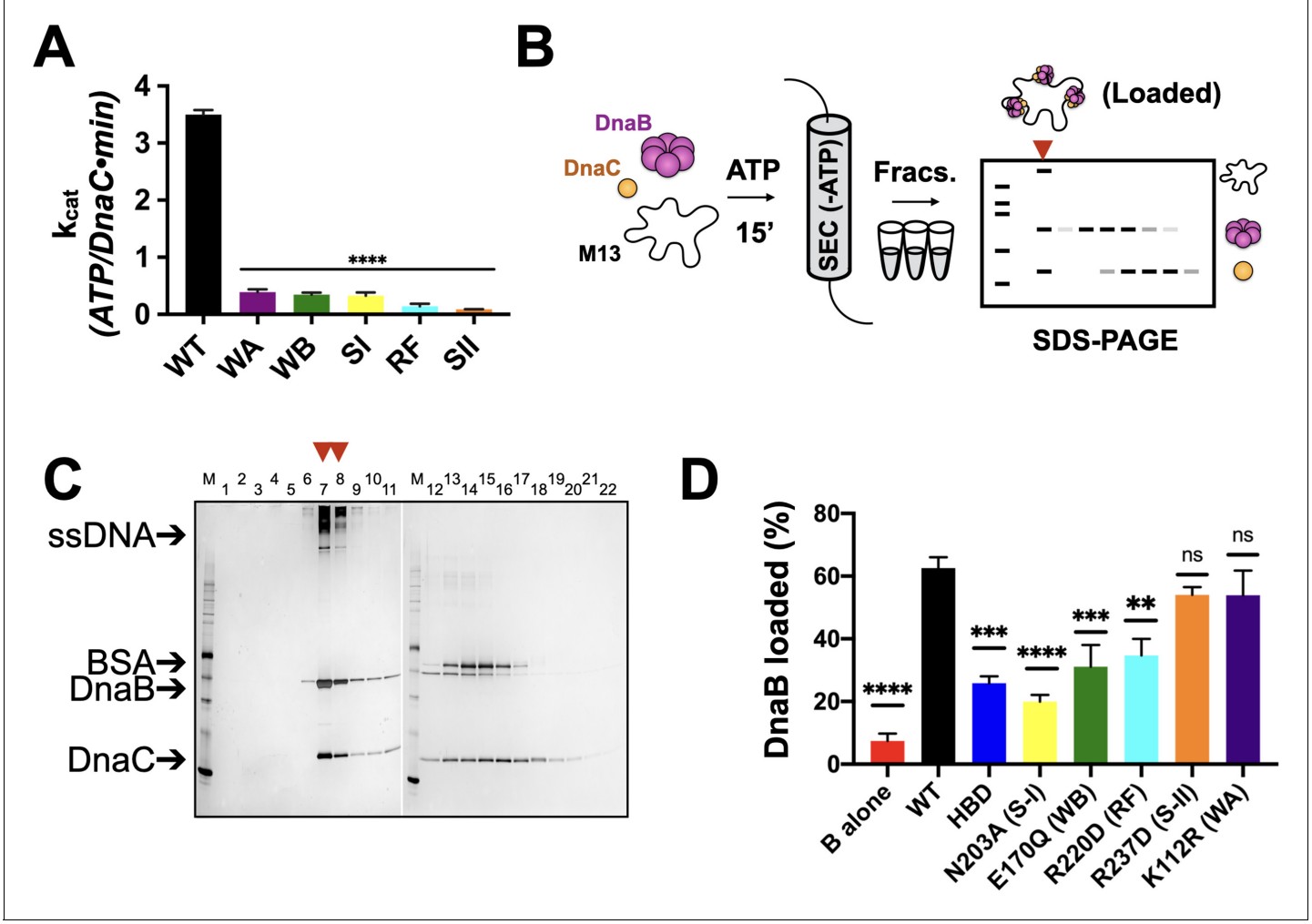

**Figure 2.** Effects of AAA+ motif mutations on DnaC ATPase and DnaB loading activities. (A) Comparison of ATPase activity between wild-type DnaC and Walker-A (WA), Walker-B, Sensor I (WB), Arg-finger (RF), Sensor I (SI), and Sensor II (SII) mutants. Assays were carried out in the presence of ssDNA and an ATPase-defective mutant (KAEA) of DnaB. (B) Schematic of DnaB loading assay. Nucleotide is generally omitted from the running buffer for the size-exclusion column. (C) Silver-stained sodium dodecyl sulphate-polyacrylamide gel electrophoresis (SDS-PAGE) analysis of fractions (lane numbers) from a representative loading reaction containing wild-type DnaB, DnaC, and M13 ssDNA that has been passed over a sizing column. Lanes where DnaB co-migrates with DNA (red arrowheads) indicate loading. Some DnaC remains bound to loaded DnaB under these conditions. 'M' – sizing marker. (D) Comparison of loading efficiencies for wild-type DnaB in the presence of different DnaC ATPase mutants. HBD – helicase-binding domain of DnaC. Error bars on this and all other bar graphs reflect the standard deviation in loading from three independent experimental replicates. Asterisks on this and all other bar graphs correspond to p-values from one-way ANOVA tests of all mutants against WT DnaC: ****$p<0.0001$; ***$p<0.001$; **$p<0.01$; *$p<0.05$; 'ns' – not significant.

The online version of this article includes the following source data and figure supplement(s) for figure 2:

**Source data 1.** Image of full WT loading assay gel and ATPase data.

**Figure supplement 1.** SDS-PAGE analysis of sizing-column fractions from DnaB loading reactions containing.

**Figure supplement 1—source data 1.** Images of full loading assay gels shown in *Figure 2—figure supplement 1*.

commensurate with or in slight excess of the HBD, whereas the Sensor II and Walker-A mutants were effectively wild-type (*Figure 2D*, *Figure 2—figure supplement 1D–G*). Collectively, these results demonstrate that the AAA+ domain and the ATP-hydrolysis cycle of DnaC increase the efficiency of helicase loading. They also show that, despite being required for ATPase function overall, different AAA+ active site residues contribute to the efficiency of DnaB deposition by varying degrees.

## Helicase loading is aided by DnaB's ATPase activity and results in the topological entrapment of ssDNA

Prior structural work has indicated that DnaC-associated DnaB switches from an ADP to an ATP state upon binding to DNA, and that the engagement of nucleic acid triggers DnaB to adopt a cracked-ring state that remains topologically linked around its DNA substrate through domain-swapping interactions (*Arias-Palomo et al., 2019*). To better define the extent to which the ATPase activity of DnaB is important for loading, we assessed the ability of wild-type DnaC to load either native DnaB or a double Walker-A/B ATPase-defective mutant (DnaB^KAEA) of the helicase using the M13 association assay. The resultant data show that the ATPase-deficient DnaB is loaded by DnaC, but at a reduced level (~50%) compared to the wild-type helicase (*Figure 3A*, *Figure 3—figure supplement 1A*). We next increased the concentration of salt in the sizing-column running buffer used to separate DNA-loaded and DNA-free DnaB. A majority of the loaded DnaB remained associated with ssDNA when NaCl was increased from 100 to 250 mM (*Figure 3B*, *Figure 3—figure supplement 1B*). These results demonstrate that the ability of DnaB to hydrolyze nucleotides is not critical for loading, but does aid the efficiency of the deposition reaction, and that once loaded, DnaB persists in a salt-stable complex with ssDNA. This latter finding indicates that the topologically closed-ring conformation observed structurally for the DnaBC complex on DNA persists in solution.

## DnaC can promote helicase unloading

Wild-type DnaC is known to remain associated with DnaB following loading, with different studies reporting different stoichiometries for the amount of DnaC that remains bound to helicase after deposition (*Davey et al., 2002*; *Fang et al., 1999*; *Makowska-Grzyska and Kaguni, 2010*). Our steady-state ATPase assays show that DnaC continuously hydrolyzes nucleotides in the presence of DnaB and DNA, indicating that the loader can cycle from a product state back to an ATP-bound configuration. This behavior raised the possibility that ADP-bound DnaC might be capable of reassociating with ATP and reopening a DnaB ring to promote helicase unloading. To test this idea, we repeated our M13 ssDNA-association reactions with wild-type DnaC, but now using sizing columns pre-equilibrated with ATP or ADP (*Figure 3C*). The presence of ADP in the column buffer still resulted in efficient DnaB loading. By contrast, the inclusion of ATP led to a near-complete loss of DnaB that remained associated with M13 DNA (*Figure 3D*, *Figure 3—figure supplement 1C–D*), demonstrating that DnaC can unload DnaB in an ATP-dependent manner. To determine whether the ATPase activity of DnaB might also play a role in unloading, we assessed the effect of ATP in the column running buffer using the DnaB^KAEA mutant. Interestingly, substantially more ATPase-deficient DnaB remained loaded on the M13 ssDNA in the presence of ATP compared to wild-type DnaB (*Figure 3—figure supplement 1E–F*). This finding shows that the ATPase activity of DnaB is important for its unloading from DNA, as well as for loading.

## Helicase activation is modulated by DnaC ATPase activity

In addition to controlling DnaB loading, DnaC has been shown to stimulate the unwinding of forked DNA substrates by the helicase (*Arias-Palomo et al., 2013*). Unlike closed-circular M13 ssDNA, DnaB can thread onto free ssDNA ends to promote strand separation of downstream duplexes without assistance (*Jezewska et al., 1998*); however, unwinding is inefficient unless DnaB is present in large excess or DnaC is included (*Arias-Palomo et al., 2013*; *Galletto et al., 2004a*; *Kaplan, 2000*; *Kaplan and Steitz, 1999*). To define the role of DnaC ATPase in promoting DnaB helicase activity, we conducted DNA unwinding assays with our panel of mutants using a forked substrate bearing a Cy3 fluorophore quenched with BHQ2 at the duplex end. When present at only a twofold molar excess over the forked substrate, DnaB was observed to unwind ~10% of the DNA (*Figure 4A*). To determine whether DnaB was inactive in this assay because a fraction of the helicase associates with the 3' tail of the fork and impedes access to the 5' tail, we blocked the 3'-end of the fork with a biotin-neutravidin complex and repeated the assay. This block resulted in a higher proportion of observed fork unwinding, but the total amount of unwound product (~45%) was still less than that seen when DnaC was present (~80%) (*Figure 4A*). Interestingly, the level of stimulation afforded by DnaC is comparable whether a 3'-end block is present or not (compare 'B + C' curves in *Figure 4A* vs. *Figure 4B*). These results indicate that 3' DNA ends can either sequester a proportion of DnaB or fold back against the duplex in a manner that can allow DnaB to translocate over the

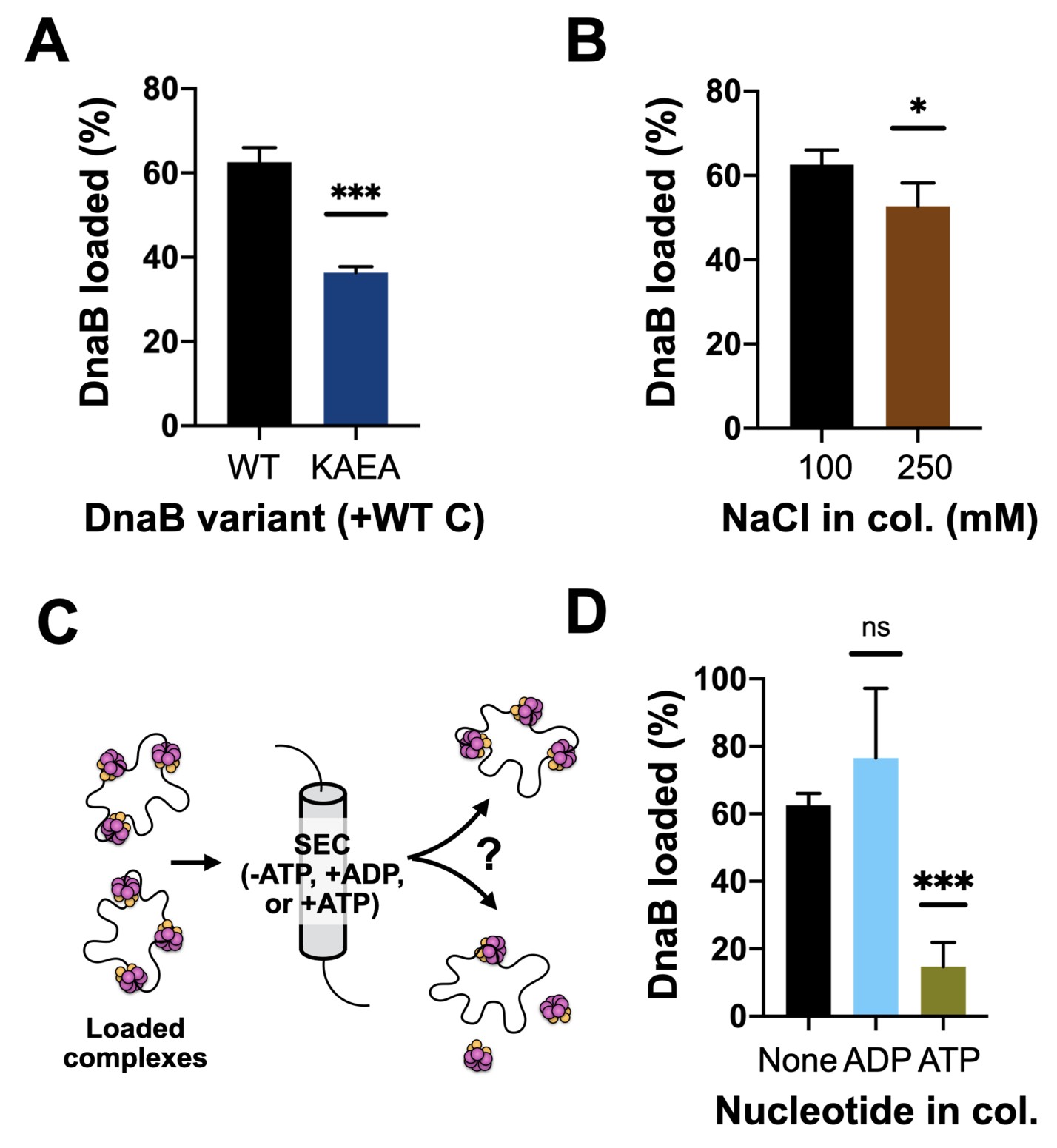

**Figure 3.** DnaB loading is impacted by helicase ATPase status and can be reversed by DnaC. (**A**) DnaB loading is compromised in an ATPase-deficient mutant of DnaB. (**B**) DnaB loaded on M13 ssDNA was largely salt resistant, consistent with structural studies showing that the helicase ring topologically closes around DNA in the presence of DnaC. (**C**) Schematic of helicase unloading assay in which different nucleotides are added to the column running buffer. (**D**) DnaC can actively unload DNA in an ATP-dependent manner. The inclusion of ATP in the column running buffer reverses stable DnaB loading, whereas ADP does not.

*Figure 3 continued on next page*

*Figure 3 continued*

The online version of this article includes the following source data and figure supplement(s) for figure 3:

**Source data 1.** Images and quantification of full loading assay gels shown in *Figure 3*.
**Figure supplement 1.** Representative loading assay gels for DnaC AAA+ mutants.
**Figure supplement 1—source data 1.** Images and quantification of full loading assay gels shown in *Figure 3—figure supplement 1*.

---

three strands without unwinding the substrate (*Kaplan and O'Donnell, 2004*), but that DnaC can overcome such inactive states.

The ability of different DnaC ATPase mutants to activate helicase unwinding was tested next. As seen previously (*Arias-Palomo et al., 2013*), the isolated HBD of DnaC readily stimulates DnaB activity on an unblocked fork, at a level close to that of the full-length loading factor (*Figure 4B*). By comparison, the DnaC ATPase mutants were much less capable in their response (*Figure 4C*, *Figure 4—figure supplement 1*). For example, the Walker-A mutant showed ~40–50% the stimulatory activity of wild-type DnaC, whereas the charge-reversed Arg-finger mutant failed to stimulate DnaB beyond the basal level of the helicase alone (other mutants were intermediate in their response, with ~25–30% the activity of the native loader). There was no clear trend between the degree to which a particular mutation disrupted both DnaC ATPase activity and the stimulation of DnaB unwinding. However, the one mutant that would be expected to interfere with ATP binding (Walker-A) evinced the highest stimulatory effect, while the mutant that would be most expected to impede inter-subunit communication (Arg-finger) was the most severely compromised. This behavior suggests that the capacity of adjoining DnaC protomers to interact with one another in a nucleotide-dependent manner can influence the extent to which the loader can stimulate DNA unwinding by DnaB.

## Multiple sensors link ssDNA binding to DnaC ATPase activity

Although AAA+ ATPase activity is often tightly coupled to interactions with client substrates, the mechanisms behind this linkage are highly varied and frequently unclear. Recent structural studies of

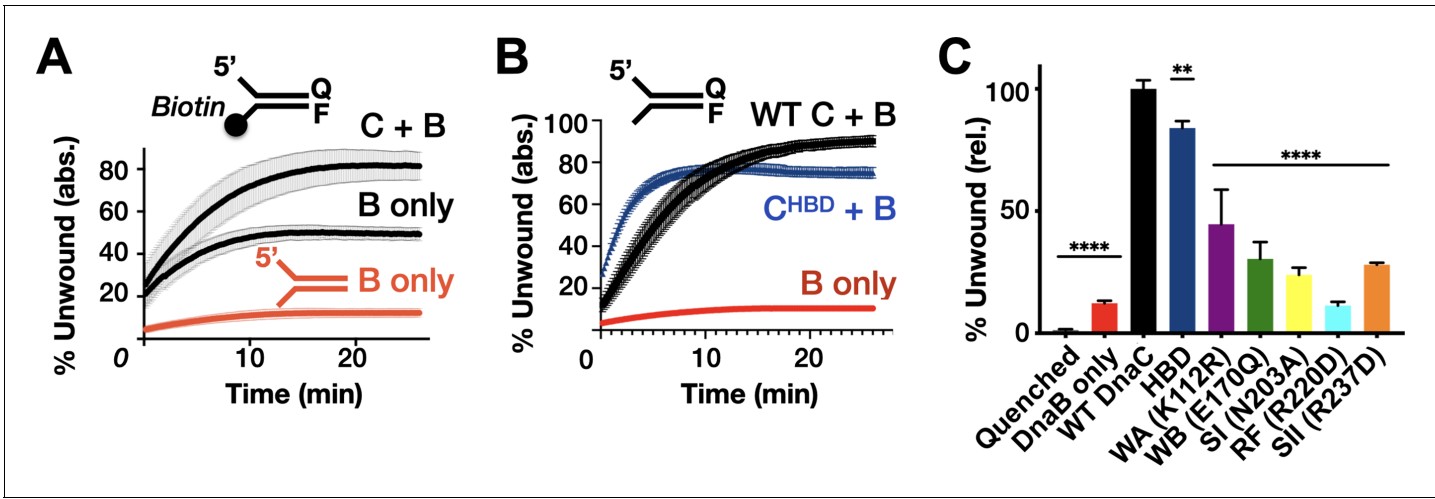

**Figure 4.** Effect of DnaC ATPase status and 3'-end accessibility on the stimulation of DNA unwinding by DnaB. (A) The addition of a 3' biotin-end block to a DNA fork partially stimulates DnaB-dependent fork unwinding compared to when DnaC is present. 'F' and 'Q' denote the positions of the fluorophore and quencher on the DNA substrates. (B) The isolated DnaC HBD readily stimulates unwinding of an unblocked fork compared to wild-type DnaC. (C) The ATPase activity of DnaC is necessary for stimulating fork unwinding by DnaB. For (A) and (B), the absolute percent of DNA strand separation as measured by a gain in fluorescence intensity is plotted as a function of time. For (C), the percent unwound shown is relative to wild-type DnaC. Error bars in this and all other unwinding curves reflect the standard deviation from different loads of the same reaction in a single experiment.

The online version of this article includes the following source data and figure supplement(s) for figure 4:

**Source data 1.** Raw unwinding data for biotinylated fork and DnaC HBD and quantification of total % unwound.
**Figure supplement 1.** Effect of DnaC ATPase status on the stimulation of DnaB unwinding activity.
**Figure supplement 1—source data 1.** Raw unwinding data of DnaC AAA+ mutants.

DnaC bound to DnaB and ssDNA noted three conserved amino acids—Phe146, Ser177, and Tyr179—which associate with the nucleic acid strand (*Figures 1* and *5A*; *Arias-Palomo et al., 2019*). Mutation of these residues blocks ssDNA binding and DNA-stimulated ATPase activity, as well as the loading of DnaB onto M13 DNA (*Arias-Palomo et al., 2019*). Their alteration also interferes with the ability of DnaC to stimulate DNA unwinding by DnaB (*Figure 5—figure supplement 1A–B*).

In the course of our phylogenetic analyses of DnaC/I-family proteins, we noted another conserved residue, Lys143, that is positioned one helical turn upstream of Phe146 (*Figures 1* and *5A*). Inspection of the DnaBC•ssDNA structure shows that this amino acid is proximal to the substrate phosphodiester backbone in a subset of DnaC protomers, forming a small patch with its Phe146, Ser177, and Tyr179 counterparts that grasps the nucleic acid substrate (*Figure 5B*). Substitution of Lys143 with glutamic acid leads to a complete loss of DNA/DnaB-stimulated ATPase

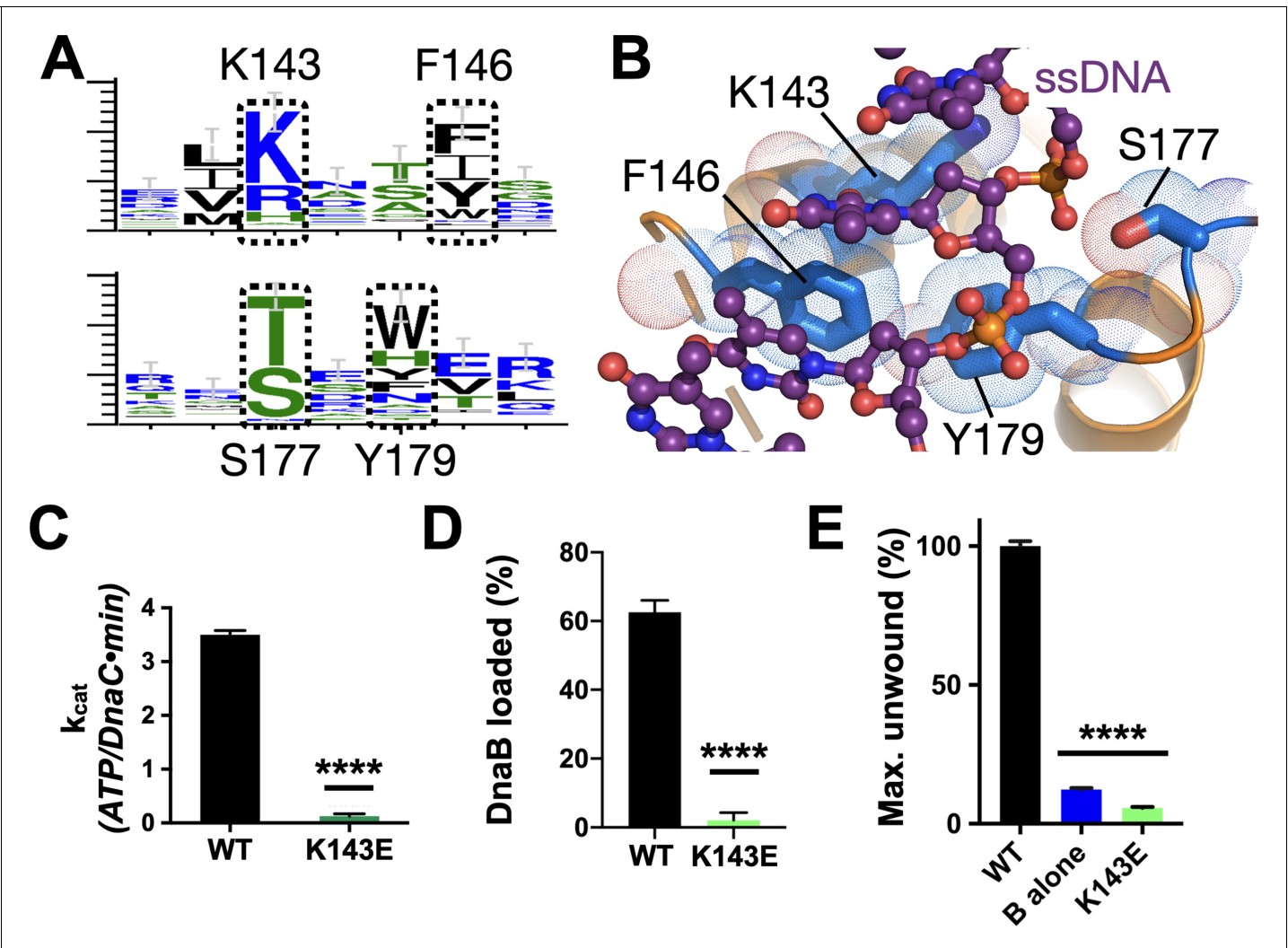

**Figure 5.** Effect of DNA binding on DnaC function. (**A**) WebLogos (*Schneider and Stephens, 1990*) showing the conservation of the Lys143/Phe146 and Ser177/Tyr179 regions of DnaC. (**B**) Ribbon and stick representation showing how ssDNA binds to a tetrad of residues in DnaC (*Arias-Palomo et al., 2019*). (**C–E**) Bar graphs showing that Lys143 of DnaC is essential to (**C**) DnaC ATPase function, (**D**) DnaB loading, and (**E**) stimulation of DNA fork unwinding by DnaB.

The online version of this article includes the following source data and figure supplement(s) for figure 5:

**Source data 1.** Raw ATP hydrolysis, helicase loading, and unwinding data for DnaC K143E.

**Figure supplement 1.** DnaC residues seen to engage ssDNA structurally are important for ssDNA binding, helicase loading, and stimulating DNA unwinding by DnaB.

**Figure supplement 1—source data 1.** Raw unwinding data for DnaC tetrad mutants and DNA binding and helicase loading data for DnaC K143E.

activity by DnaC and strongly impaired DNA binding (*Figure 5C*, *Figure 5—figure supplement 1C*). The K143E mutant also proved incapable of loading DnaB onto M13 DNA or activating DNA strand separation by the helicase (*Figure 5D–E*, *Figure 5—figure supplement 1D–E*). These findings demonstrate that, as with other conserved amino acids in its local neighborhood, Lys143 is a part of the mechanism used by DnaC to sense ssDNA.

A second conserved element in DnaC/I-family proteins revealed by sequence homology is the presence of an invariant arginine, Arg216, one helical turn upstream of the Arg-finger (*Figures 1* and *6A*). To more deeply probe the role of Arg216, we mutated the amino acid to both aspartate (charge reversal) and alanine (charge ablation) and assessed the function of these constructs biochemically. The R216D mutant proved unable to hydrolyze ATP, did not stimulate DNA unwinding by DnaB, and could only support DnaB loading at a level comparable to the Sensor I mutant (*Figure 6B–D*, *Figure 6—figure supplement 1A–B*). By contrast, DnaC$^{R216A}$ showed a moderate capacity to support helicase loading and stimulated DNA unwinding by DnaB with wild-type efficiency (*Figure 6C–D*, *Figure 6—figure supplement 1A,C*). Surprisingly, the R216A construct actually exhibited slightly higher ATPase activity than native DnaC and additionally required only DnaB (and not ssDNA) for ATPase function (*Figure 6B*). The alanine mutant also proved capable of counteracting the strong inhibitory block placed on DnaC ATPase activity in the context of a K143E substitution (*Figure 6E*), although it was unable to restore the activation of DNA unwinding by DnaB in such a context (*Figure 6—figure supplement 1D*). These results reveal that Arg216 is as a key coupling element in DnaC, repressing nucleotide turnover in the presence of DnaB until ssDNA also associates with the complex.

## A replication restart mutant disrupts DNA/ATPase coupling in DnaC

In addition to supporting helicase loading during replication initiation, DnaC also participates in replication restart (*Schekman et al., 1975*; *Wickner and Hurwitz, 1974*). Numerous DnaC variants have been isolated as suppressors for the loss of critical replication restart proteins in genetic screens (*Sandler et al., 1996*); one is *dnaC810*, which results from the substitution of Glu176 with glycine. We noted that Glu176, though not particularly well-conserved, sits proximal to the DNA-binding residues Ser177 and Tyr179, in a loop adjoining the Walker-B element that undergoes a conformational switch between DNA-free and -bound states of DnaC (*Figure 1C*, *Figure 6—figure supplement 1E*).

To better understand the physical consequences of the *dnaC810* mutation, we purified an E176G version of the protein and assessed its impact on DnaC function in vitro. The variant proved capable of supporting both DnaB loading and the activation of fork unwinding by the helicase, at levels only moderately decreased from native DnaC (*Figure 7A–B*, *Figure 6—figure supplement 1F–G*). DnaC$^{E176G}$ was also capable of supporting the ssDNA- and DnaB-dependent ATPase function of the loader (*Figure 7C*); interestingly, the activity of the mutant loader exceeded that of wild-type protein by ~50%. In addition, the substitution allowed DnaC to hydrolyze ATP when bound to DnaB alone, although at approximately half the rate as when DNA was present. These data demonstrate that, like the R216A (Arg-coupler) substitution (*Figure 6B*), the mutation present in the *dnac810* strain has gained the ability to act on DnaB in a DNA-independent manner.

The similar biochemical behaviors of the DnaC810 mutant to the R216A variant raised the question as to whether the Arg-coupler is important for DnaC to bypass the need for replication restart factors in loading DnaB onto SSB-bound DNA. To address this question, we tested the ability of DnaC$^{R216A}$ to support DNA unwinding by DnaB on an SSB-coated fork substrate (*Figure 7D*, *Figure 6—figure supplement 1H*). Wild-type DnaC only weakly supported fork unwinding but was modestly stimulated by the presence of the restart factor, PriC. By contrast, DnaC810 supported a high level of fork unwinding even in the absence of PriC. Interestingly, DnaC$^{R216A}$ displayed an even greater ability to stimulate DnaB-mediated unwinding of the SSB-bound substrate than DnaC810 when PriC was absent. These data demonstrate that the Arg-coupler plays a role in preventing DnaC from activating DnaB unless restart factors are present.

## Discussion

Cells rely on ATP-dependent protein/protein and protein/nucleic-acid complexes to support a diverse array of critical biochemical processes. A group of enzymes known as AAA+ ATPases are

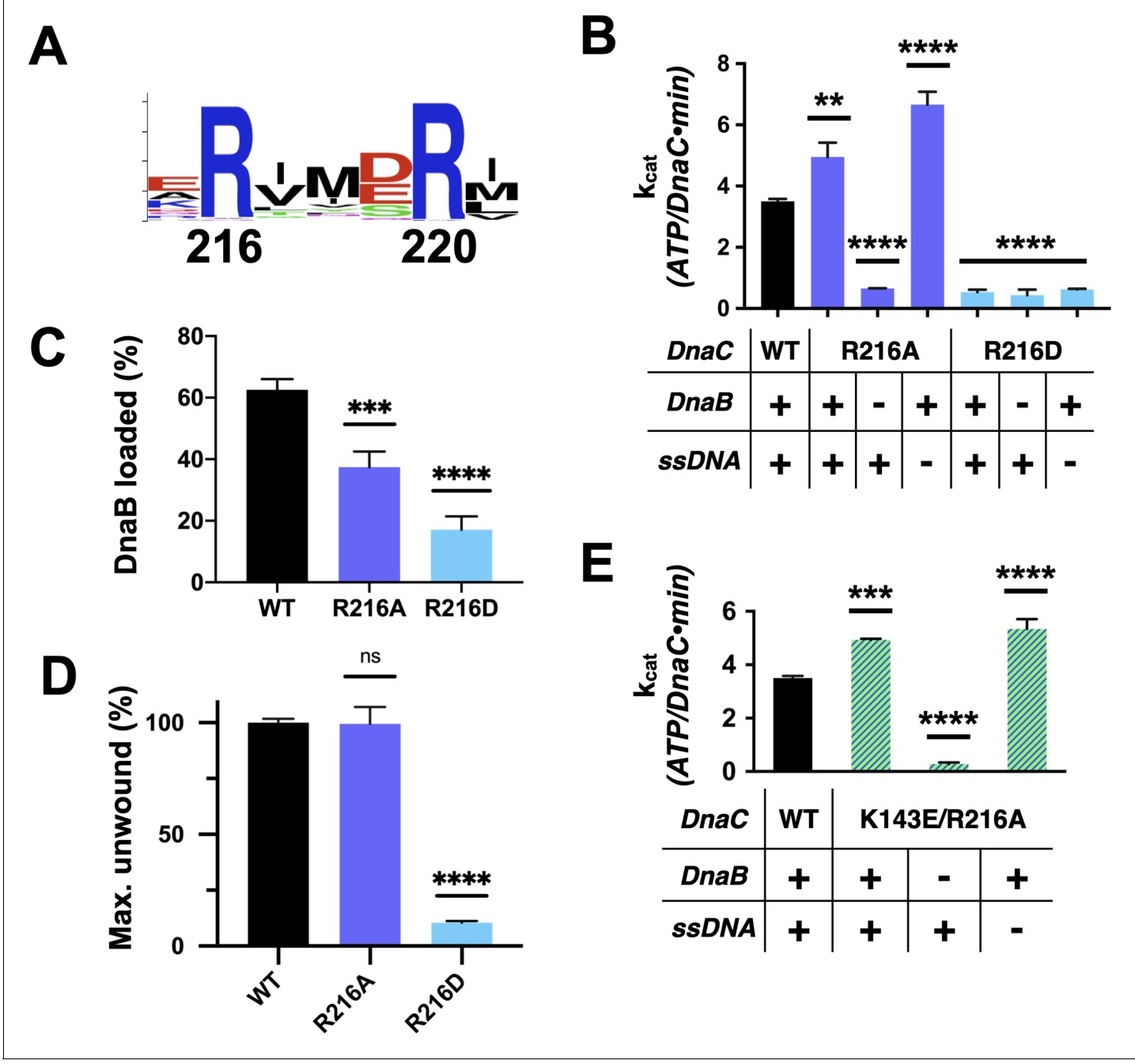

**Figure 6.** Functional effect of a second conserved arginine adjacent to the Arg-finger on DnaC. (A) WebLogo (*Schneider and Stephens, 1990*) showing the conservation of Arg216, which sits four amino acids away from the DnaC Arg-finger (Arg220). (B–D) Bar graphs showing the effect of R216A and R216E mutants on (B) ssDNA- and DnaB-dependent stimulation of DnaC ATPase activity, (C) DnaB loading by DnaC, and (D) DnaC stimulation of DNA fork unwinding by DnaB. (E) Bar graph showing that the R216A mutation can override the loss of DnaC ATPase activity imparted by the disruption of Lys143 when DnaB is present.

The online version of this article includes the following source data and figure supplement(s) for figure 6:

**Source data 1.** Raw data on unwinding, loading, and DNA and DnaB stimulated ATP hydrolysis by the arginine coupler in DnaC.

**Figure supplement 1.** Effect of coupler amino acids on DnaC function.

**Figure supplement 1—source data 1.** Raw data on effects of coupler residues on DnaC function.

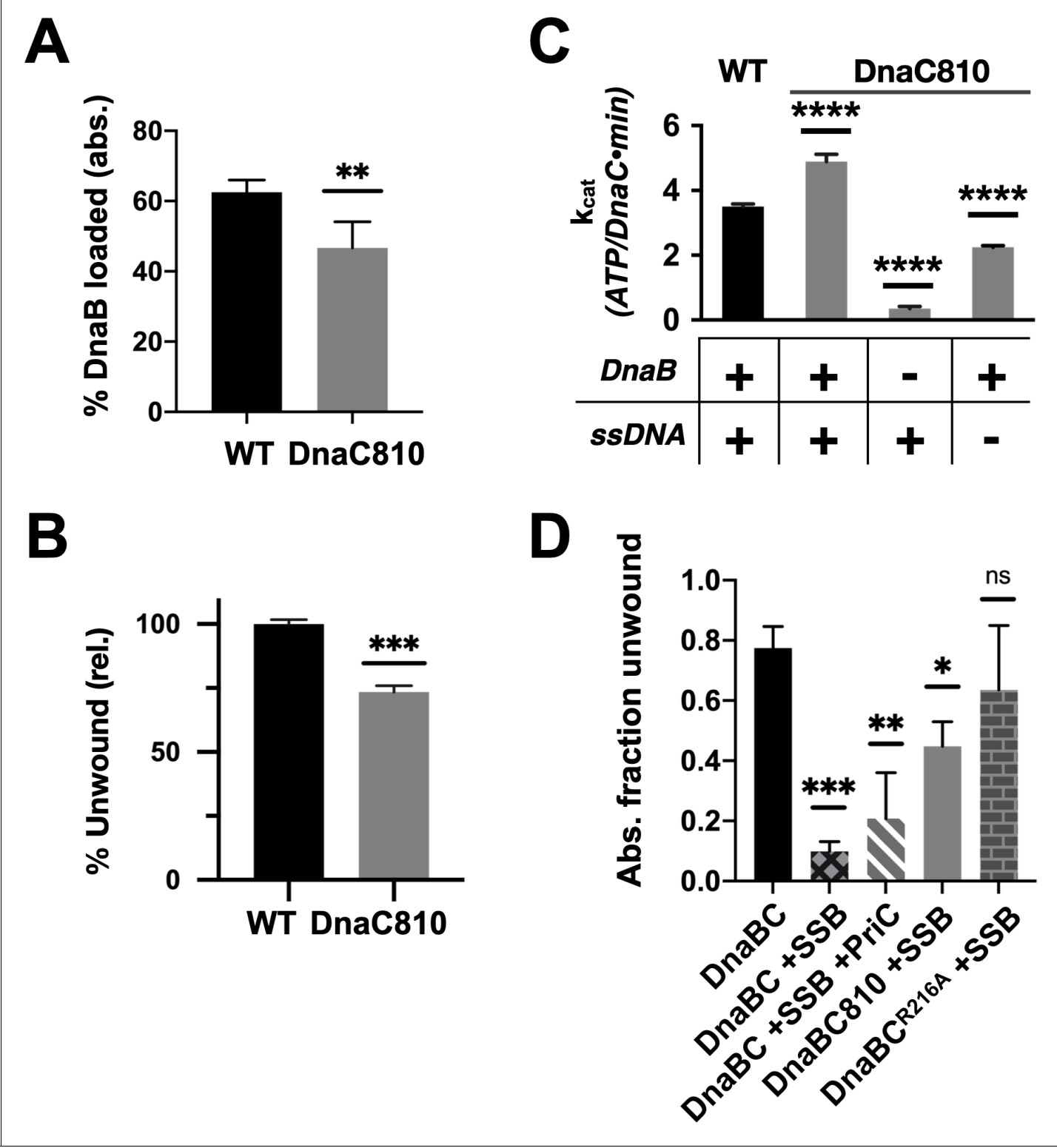

**Figure 7.** Loss of ssDNA dependency on DnaC ATPase activity correlates with an ability to bypass a need for the replication restart factor, PriC, in unwinding SSB-coated DNA. (A–C) The DnaC810 replication restart bypass mutation phenocopies a loss of the Arg-coupler. Bar graphs showing the effect of the DnaC810 E176G mutation on (A) percent of DnaB loading (absolute values), (B) percent of DnaB-dependent DNA fork unwinding (relative to wild-type DnaC), and (C) ssDNA- and DnaB-stimulated ATPase activity of the loader. (D) The Arg-coupler mutant of DnaC promotes the DnaB-mediated unwinding of an SSB-coated fork in a PriC-independent manner. Error bars reflect the standard deviation from three different experiments.

*Figure 7 continued on next page*

*Figure 7 continued*

The online version of this article includes the following source data for figure 7:

**Source data 1.** Raw data on effect of DnaC 810 mutant on DnaC function.

exemplars of such molecular 'machines,' participating in events ranging from DNA replication and gene expression to proteolytic homeostasis and vesicle trafficking (*Neuwald et al., 1999b*). How AAA+ enzymes couple ATP turnover to movement and function is a broadly studied mechanistic question. Although the appropriate engagement of client substrates is clearly important for ATPase coupling, the elements responsible for these interactions are typically far removed (20–30 Å) from the nucleotide-hydrolysis center. The structural linkage elements that relay information over a distance between client occupancy and ATPase status are rarely known.

Here, we have used structure-guided mutagenesis and biochemical analyses to dissect how DnaC, a bacterial AAA+ ATPase that supports the deposition of the replicative DnaB helicase onto DNA, coordinates its ATPase cycle with helicase loading and activation. The opening of the DnaB ring and its subsequent engagement with DNA can occur independently of ATP hydrolysis by DnaC (*Figure 2*), a finding in accord with previous data showing that neither ATP binding nor the DnaC ATPase domains are required for DnaB loading in vitro (*Arias-Palomo et al., 2013*; *Davey et al., 2002*). However, ATP turnover does improve the efficiency of helicase loading (*Figure 2*), an observation in agreement both with early work of *Wahles et al., 1988* (which reported that an ATP-hydrolysis step is important in the binding of the DnaBC complex to DNA) and with a study showing that the replication factor C (RFC) clamp loader becomes compromised for DNA binding when its D or E subunit bears a mutated Arg-finger (*Johnson et al., 2006*). Interestingly, the ability to hydrolyze nucleotides also has a potentiating effect on the capacity of DnaC to activate DNA unwinding by DnaB (*Figure 4*). Indeed, hydrolysis-defective DnaC mutants are less capable of activating DnaB than an ATP-binding (Walker-A, K112R) mutant. Moreover, all ATPase mutants are substantially compromised for activating DNA unwinding compared to wild-type DnaC or the helicase-binding domain of DnaC alone. These observations indicate that the HBD is an independent helicase-activating element regulated by its associated ATPase module and that ATP turnover by the loader is important for allowing DnaB to shift into a translocation-competent form.

Recent images of DNA-free and DNA-bound structures of the DnaBC complex have revealed that the nucleotide status of the helicase flips from an ADP state to an ATP configuration when DNA is present (*Arias-Palomo et al., 2019*). This observation suggested to us that the loading of DnaB onto DNA might be impacted by the helicase's ATPase function. In testing this idea with a double Walker-A/Walker-B (KAEA) mutant of DnaB, the ability of the helicase to interact productively with nucleotides indeed impacts DNA loading (*Figure 3A*). Although this coupling is not a strict dependency, a role for nucleotide turnover by DnaB is reminiscent of the need for ATP by the eukaryotic Mcm2-7 replicative helicase to support its loading onto DNA by the origin recognition complex (ORC) (*Coster et al., 2014*). Interestingly, we found that DnaC can also drive helicase unloading as well as loading (*Figure 3D*). In contrast to eukaryotes, where unloading of the Cdc45/Mcm2-7/GINS helicase is facilitated by ubiquitylation of MCM7 and removal of the CMG helicase by CDC48/p97 (*Deng et al., 2019*), relatively little is known about how DnaB helicases are removed from DNA following replication termination in bacteria. Additional studies will be needed to assess a potential role for DnaC in this process.

Structural work has shown that the AAA+ fold of DnaC associates with DNA directly (*Arias-Palomo et al., 2019*). Three residues—Phe146, Ser177, and Tyr179—form a small DNA-binding locus on the part of the DnaC ATPase domain that faces the interior pore of the dodecameric DnaBC complex; mutation of these residues ablates DnaB loading by DnaC (*Arias-Palomo et al., 2019*). In examining this region further, we found a fourth amino acid, Lys143, that also participates in binding DNA. Lys143 is generally a lysine or arginine in DnaC/DnaI homologs, and a prior genetic screen *Hupert-Kocurek et al., 2007* noted that the integrity of this residue is important for *E. coli* viability. Here, we have shown that Lys143 is important for supporting the ATPase and DnaB loading activities of *E. coli* DnaC (*Figure 5*). The inability of a DnaC$^{K143E}$ mutant to facilitate DnaB loading onto ssDNA—even though it has an intact HBD and stably binds DnaB—suggests that the alteration allows the loader to maintain a persistent, open-ring state of the helicase. In addition, we found that

Lys143 and other amino acids in the DNA-binding patch are critical for DnaC to stimulate fork unwinding by DnaB (*Figure 5E*, *Figure 5—figure supplement 1A–B*). Biochemical studies show that disruption of this residue and other amino acids in the region block DNA binding (*Figure 5—figure supplement 1C*; *Arias-Palomo et al., 2019*), thereby short-circuiting the ability of the nucleic acid substrate to stimulate ATP hydrolysis by DnaC when the loader is bound to DnaB. Overall, the integrity of the DNA-binding patch on DnaC supports the ability of ssDNA to activate all known DnaC functions, even though it is situated over 20 Å away from the DnaC ATPase center.

A long-standing mechanistic question has been to understand why DnaB is required for stimulating ATP turnover by DnaC, yet is incapable of supporting this activity unless ssDNA is also present (*Davey et al., 2002*). Unlike a majority of AAA+ ATPases, *E. coli* DnaC is a monomer on its own (*Galletto et al., 2004b*) and forms stable oligomeric interactions only when templated by a pre-existing DnaB hexamer (*Kobori and Kornberg, 1982*). Formation of a higher-order oligomer state is generally a prerequisite for AAA+ ATPase function, as the active site of one subunit typically depends on the contribution of an arginine or lysine from a partner subunit to support nucleotide turnover (*Neuwald, 1999a*; *Neuwald et al., 1999b*). Prior work has shown that mutation of the so-termed 'Arg-finger' residue to alanine in *E. coli* DnaC does not interfere with the ability of the loader to deliver DnaB to *oriC*, but does interfere with DnaC-dependent DNA replication in vitro and DnaC function in vivo (*Makowska-Grzyska and Kaguni, 2010*). Here we have shown that this amino acid is critical for DnaC ATPase activity (*Figure 2A*) and that while it is only moderately important for DnaB loading, its presence is necessary to activate DNA unwinding by the helicase (*Figures 2D* and *4C*).

Interestingly, DnaC possesses a second arginine (Arg216) just upstream of its Arg-finger (Arg220). This amino acid is essentially invariant among DnaC/DnaI homologs (*Figure 6A*). Prior studies have shown that the integrity of this residue is unnecessary for DnaC to bind ATP or associate with DnaB but is important for supporting replication in vitro and for cell viability (*Makowska-Grzyska and Kaguni, 2010*). Here, we have shown that Arg216 is actually an essential element for coupling substrate DNA binding to loader ATPase activity. Mutation of the amino acid to aspartate severely impairs DnaC's ATPase activity and its ability to mobilize and activate DnaB on DNA. However, substitution with alanine not only increases the rate of ATP hydrolysis by DnaC, but also allows nucleotide turnover to occur even in the absence of DNA (*Figure 6B*). The DnaC^R216A mutant also overrides the ATPase defect imparted by a DNA-binding mutant (e.g., K143E), yet does not degrade the ability of DnaC to assist with either DnaB loading or the stimulation of DnaB-mediated fork unwinding. Inspection of a DNA-free DnaBC structure reveals that Arg216 is positioned where the nucleophilic water would be expected to reside prior to ATP hydrolysis; in the presence of ssDNA, this amino acid shifts away to allow access to the reactive γ-PO$_4$ (*Figure 8A–B*). Thus, Arg216 acts not just as a substrate/ATPase coupler, but also as a steric governor, preventing the DnaC ATPase from firing when oligomerized by DnaB to prevent futile cycling until substrate DNA is bound. The action of the Arg-coupler explains how the ATPase activity of DnaC allosterically responds to the binding of ssDNA at a region distal to the active site. The binding of ssDNA to the tetrad residues elicits a subtle rocking and compression motion between neighboring DnaC protomers (*Arias-Palomo et al., 2019*); this action leads the Arg-coupler to shift out of the active site, releasing its repressive action on ATPase function (*Figure 1—video 1*). In this regard, the Arg-coupler acts in a manner akin to the 'glutamate switch', a residue found in many AAA+ ATPases that controls the conformation of the catalytic glutamate in response to ligand binding (*Zhang and Wigley, 2008*). Interestingly, a polar amino acid at the position of the glutamate switch is present in *E. coli* DnaC (Thr134), but it does not make ideal hydrogen-bonding interactions with the catalytic glutamate (Glu170) in the repressed (DNA-free) ATPase state (*Figure 8—figure supplement 1A*). Instead, Thr134 forms a contact with an acidic amino acid (Asp 189), which is in turn hydrogen-bonded to the Arg-finger. Neither the glutamate switch threonine nor its acidic anchor residue is particularly well conserved among DnaC/DnaI homologs, although these two amino-acid positions are typically occupied by polar residues (*Figure 8—figure supplement 1B–C*). Whether the glutamate switch has been repurposed to control the Arg-finger instead of the catalytic glutamate in bacterial helicase loaders remains to be determined.

In examining the activity of the Arg-coupler mutant, we also probed the activity of a DnaC mutant (E176G) encoded by the *dnaC810* allele. Purified DnaC810 has been shown to load DnaB on SSB-coated ssDNA and on D-loops, bypassing the need for the PriA replication restart protein, and it has been proposed that the DnaB loading activity of this mutant is mediated by a cryptic DNA-binding

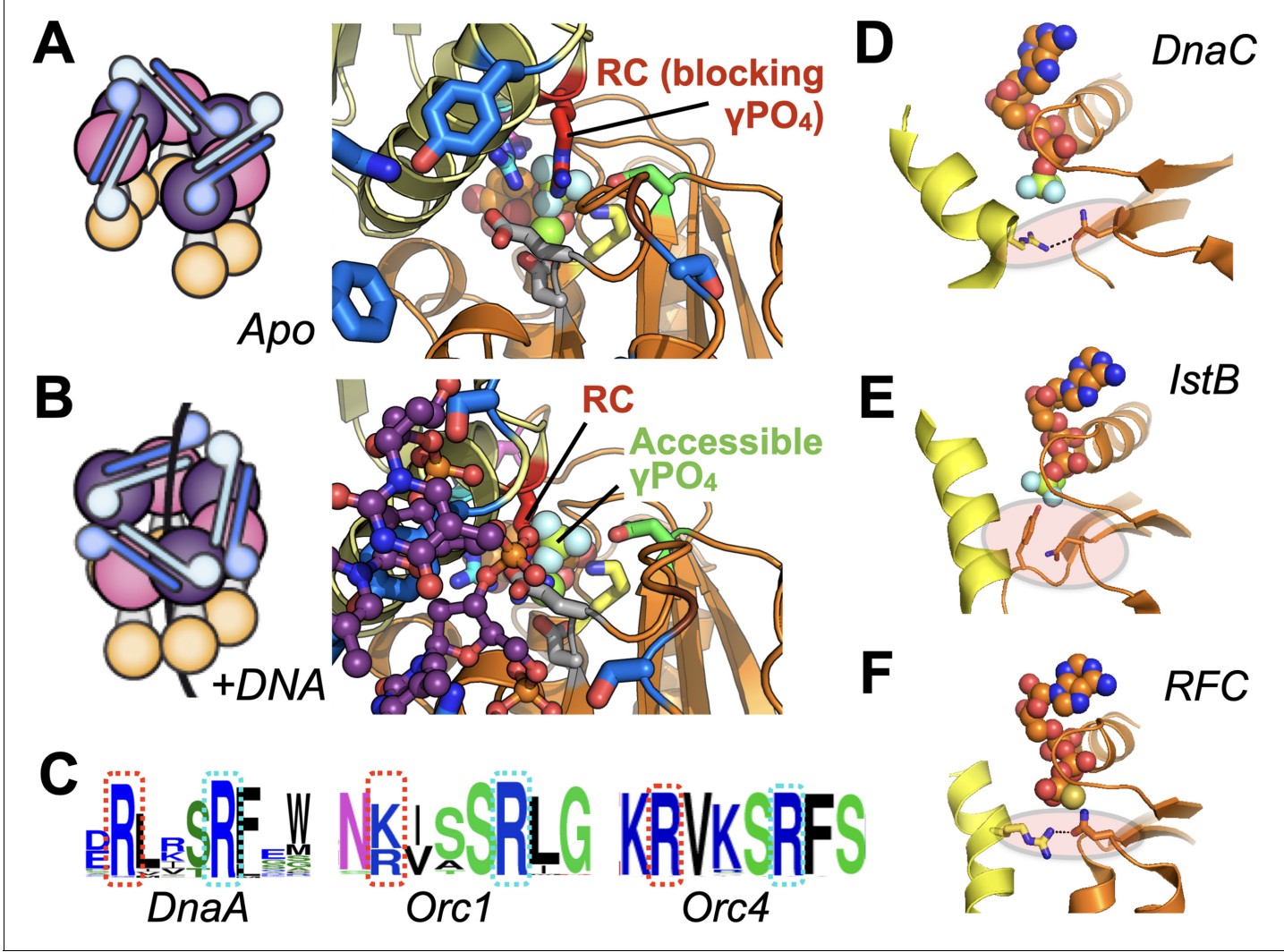

**Figure 8.** Molecular mechanism of the DnaC Arg-coupler and the presence of potential functional analogs in related switch-type AAA+ ATPases. (A,B) Schematics and structures showing how the conformational status of the Arg-coupler switches between apo and DNA-bound states to respectively occlude or allow access to the γ-PO4 moiety of nucleotide in the DnaC ATPase site. (C) WebLogo (*Schneider and Stephens, 1990*) showing an analog of Arg216 (red dotted box) is present next to the Arg-finger (cyan dotted box) of DnaA, Orc1, and Orc4, which sits four amino acids away from the DnaC Arg-finger. (D–F) The Sensor I residue of IstB and Replication Factor C (RFC, chain C in orange and chain D in yellow) is engaged by a conserved side chain in unstimulated ATPase states. DnaC PDB 6QEL; IstB PDB 5BQ5; RFC PDB 1SXJ.

The online version of this article includes the following figure supplement(s) for figure 8:

**Figure supplement 1.** The glutamate-switch region of DnaC.

site in DnaC (*Xu and Marians, 2000*). Inspection of the structure shows that Glu176 sits on a loop that connects one-half of the DNA-binding tetrad (Ser177 and Tyr179) to the Walker-B loop of the ATPase center (*Figure 1C*). Interestingly, DnaC810 shows a similar pattern of behavior as the alanine substitution of Arg216: DnaB-dependent but ssDNA-independent ATPase activity, and near wild-type efficiency in loading DnaB onto DNA and activating the helicase for fork unwinding (*Figure 7*). However, unlike the R216A mutant, the *dnaC810* allele supports cell viability (*Makowska-Grzyska and Kaguni, 2010*; *Sandler et al., 1996*). A comparison of structures of the DnaBC complex in DNA-free and -bound states shows that the Walker-B loop on which Glu176 sits switches between two conformations (*Arias-Palomo et al., 2019*). Although a structure of DnaC810 bound to DnaB is currently unavailable, it seems likely that the substitution of the glutamic acid with glycine alters the flexibility of the Walker-B region, perhaps mimicking a DNA-bound conformational state

to relieve DnaC of the repressive action of Arg216. The finding that the DnaC810 can act in a manner similar to that of the arginine-coupler mutant suggests that tight ATPase control by the loader is necessary to properly coordinate its activity with pathways that resuscitate stalled replication forks. Understanding precisely how this action might allow DnaC$^{810}$ and the R216A mutant to act independently of replication restart factors awaits further study.

Given the widespread nature of AAA+ proteins, we were curious as to whether the coupling mechanisms we observed for DnaC might be found in other ATPase family members. Because AAA + ATPases recognize diverse types of substrates—single- and double-stranded nucleic acids, protein chains—it seemed unlikely that there would be any consensus to their substrate-recognition sites. However, DnaA, a DnaC paralog that serves to reorganize and melt origins to initiate replication in bacteria, does retain a highly conserved arginine in the same position as the Arg-coupler. Sequence analyses show that two proteins used to catalyze replicative helicase loading in eukaryotes, Orc4 and Orc1, also retain such an element (*Figure 8C*). Interestingly, these proteins all belong to the same 'initiator' subgroup, or clade, of AAA+ proteins, suggesting that the coupling arginine seen in DnaC may play a similar role in regulating substrate-dependent ATP turnover in these proteins as well. In the case of DnaA, the integrity of the equivalent of DnaC-Arg216 (Arg281 in *E. coli* DnaA) has been shown to be dispensable for ATP binding and melting of *oriC* DNA, but necessary for forming a sufficiently stable complex that can support DnaB loading by DnaC (*Felczak and Kaguni, 2004*); by comparison, mutation of the neighboring Arg-finger of DnaA, R285A, leads to a failure in unwinding *oriC* (*Kawakami et al., 2005*). These findings echo the retention of certain specific activities but an overall functional defect seen for DnaC$^{R216A}$. Interestingly, although there is no known data on the phenotypic effect of mutating the equivalent residue in human Orc4 (Arg205), mutation of the analogous residue in Orc1 (Arg666, to Trp) has been linked to Meier-Gorlin syndrome (*Guernsey et al., 2011*). It has yet to be determined biochemically whether Arg-coupler mutations in ORC might also lead to an unlinking of ATPase activity from DNA-binding status (as is seen for DnaC) or to a different functional defect.

Other proteins in the initiator clade, such as IstB (a transposition helper protein and another DnaC paralog), do not possess a clear homolog of the Arg-coupler, nor do several archetypal AAA + ATPases of other clades (e.g., clamp loader, classic, pre-Sensor I insert, etc). A review of several representative structures of these proteins nonetheless does reveal a potential common theme shared between DnaC, IstB, and clamp loaders. In addition to blocking access to the reactive γ-PO$_4$ of bound nucleotide, the Arg-coupler of DnaB-bound DnaC also engages the polar Sensor I amino acid (*Figure 8D*) and releases this interaction when DNA is bound. The Sensor I amino acid is a conserved feature of AAA+ proteins and is generally important for ATP hydrolysis (*Hattendorf and Lindquist, 2002*; *Neuwald et al., 1999b*; *Steel et al., 2000*). In the case of IstB— whose ATPase is primed by DNA binding but not actuated until it engages its client IstA transposase—a DNA-bound structure of the protein shows that the Arg-coupler is replaced with a small amino acid (e.g., alanine), but that a bulky amino acid (tyrosine) on the Walker-B loop occupies the same general position to occlude the γ-PO$_4$ moiety and remodel the Sensor I loop (*Figure 8E*; *Arias-Palomo and Berger, 2015*). Clamp loaders, whose AAA+ clade is closely related to that of the initiators, also lack the upstream Arg-coupler; however, in the presence of a client clamp—which is necessary but insufficient for stimulating ATPase activity on its own—access to the γ-PO$_4$ is again occluded by a Sensor I interaction with a positively charged residue on an adjacent helix (*Figure 8F*). A comparison of clamp loader structures reveals that an arginine or lysine in this highly conserved central helix can, upon DNA binding, swing out to release the Sensor I residue, thereby allowing the Arg-finger to access the γ-PO$_4$ (*Bowman et al., 2004*; *Gaubitz et al., 2020*; *Simonetta et al., 2009*). Interestingly, an N-terminal motif known as 'RFC box II' also has been shown in the archaeal clamp loader system to be important for linking ATP hydrolysis to clamp loading: when ablated, the loader shows elevated ATPase activity but is compromised for clamp loading, a behavior similar to that of the DnaC Arg-coupler (*Seybert et al., 2006*). An interesting distinction of initiators and clamp/helicase/transposase loaders compared to other AAA+ systems is that they are switch-like in their behavior, as opposed to acting in a more processive manner. Restricting access to γ-PO$_4$ of ATP and sequestering the Sensor I amino acid may thus serve as a general fail-safe in these systems, as it does in DnaC, to allow an active ATPase center to be assembled while avoiding premature hydrolytic events. Whether other switch-like AAA+ proteins have evolved similar regulatory mechanisms remains to be determined.

# Materials and methods

## Key resources table

| Reagent type (species) or resource | Designation | Source or reference | Identifiers | Additional information |
|---|---|---|---|---|
| Recombinant DNA reagent | pET28b | Novagen | | |
| Strain, strain background (*Escherichia coli*) | C41(DE3) | Lucigen | | Chemically competent cells |
| Strain, strain background (*Escherichia coli*) | BL21AI | Invitrogen | | Chemically competent cells |
| Peptide, recombinant protein | TEV | QB3 Macrolab (UC Berkeley) | | |
| Commercial assay or kit | Amylose Resin | NEB | | |
| Commercial assay or kit | Pyruvate kinase/lactate dehydrogenase mix | Sigma-Aldrich | | |
| Commercial assay or kit | Bio-Gel A-1.5m Resin | Bio-Rad | | |

## Protein expression and purification

6xHis-MBP-tagged *E. coli dnaC* was cloned into a pET28b-derived plasmid, while *E. coli dnaB* was cloned into pET28b without an affinity tag. All constructs were sequence verified. Wild-type proteins were expressed in strain C41 (Lucigen) by induction at mid-log phase ($OD_{600}$ ~0.4) with 1 mM iso-propyl β-D-1-thiogalactopyranoside (IPTG) at 37°C for 3 hr (DnaB) or 2.5 hr (DnaC). Following induction, cells were harvested by centrifugation and resuspended in lysis buffer supplemented with protease inhibitors (for DnaB: 20 mM 4-(2-hydroxyethyl)piperazine-1-ethanesulfonic acid potassium salt (HEPES-KOH) (pH 7.5), 500 mM NaCl, 10% glycerol, 10 mM $MgCl_2$, 0.1 mM ATP, 1 mM β-mercaptoethanol; for DnaC: 50 mM HEPES-KOH (pH 7.5), 1 M KCl, 10% glycerol, 30 mM imidazole, 10% $MgCl_2$, 0.1 mM ATP, 1 mM β-mercaptoethanol; 1 mM phenylmethylsulfonyl fluoride (PMSF), 1 μg/ml pepstatin A, and 1 μg/ml leupeptin in both). Resuspended cells were flash-frozen and stored at −80°C for later use.

For purification, cells containing expressed protein were thawed, lysed by sonication, and clarified by centrifugation. Clarified lysates containing wild-type DnaB were ammonium sulfate precipitated (30% wt/v), followed by resuspension of the pellet in 100 mM NaCl Q Buffer (100 mM NaCl, 20 mM HEPES-KOH (pH 7.5), 10% glycerol, 10 mM $MgCl_2$, 0.01 mM ATP, 1 mM β-mercaptoethanol, and protease inhibitors), and passed over a HiTrap Q HP anion exchange column (GE Healthcare). Peak fractions were pooled, concentrated in Amicon Ultra Centrifugal Filters (30,000 MWCO; Millipore), and further purified by gel filtration on a HiPrep 16/60 Sephacryl S-300 column (GE Healthcare) in a buffer containing 20 mM Tris-HCl (pH 8.5), 800 mM NaCl, 10% glycerol, 5 mM $MgCl_2$, 1 mM β-mercaptoethanol, 0.1 mM ATP, 1 mM PMSF, 1 μg/ml pepstatin A, and 1 μg/ml leupeptin. Peak fractions were then concentrated and applied to a second sizing column run in a buffer containing 100 mM NaCl rather than 800 mM NaCl.

Purification for WT DnaC from clarified lysates was performed using Ni-Sepharose (GE Healthcare) affinity resin, followed by Tobacco Etch Virus (TEV) protease incubation (to remove the affinity tag) and a second Ni-Sepharose step. The flow-through from this step was concentrated in Amicon Ultra Centrifugal Filters (10,000 MWCO; Milipore) and applied to a HiPrep 16/60 Sephacryl S-200 gel filtration column (GE Healthcare) in a buffer containing 50 mM HEPES (pH 7.5), 500 mM KCl, 10% glycerol, 10 mM $MgCl_2$, 0.1 mM ATP, 1 mM β-mercaptoethanol, 1 mM PMSF, 1 μg/ml pepstatin A, and 1 μg/ml leupeptin.

To overcome expression problems arising from toxicity, mutants of DnaB and DnaC were expressed into the periplasm. Wild-type *dnaB* and *dnaC* genes were cloned as 6xHis-MBP fusions with an N-terminal periplasmic localization signal, and point mutations were introduced into these plasmids by QuickChange site-directed mutagenesis. All constructs were sequence verified. Mutant

proteins were expressed in strain C41 (Lucigen) by induction at mid-log phase (OD$_{600}$ ~0.4) with 0.5 mM IPTG overnight at 16°C. Following induction, cells were harvested by centrifugation, resuspended into the appropriate lysis buffer for the wild-type proteins, flash-frozen, and stored at −80°C for later use. Purification protocols followed those of wild-type proteins, except that the DnaB$^{KAEA}$ double mutant was purified using sequential Ni-Sepharose (GE Healthcare) and amylose (New England Biolabs) affinity resins, followed by TEV protease incubation (to remove the affinity tag) and a second Ni-Sepharose step.

One mutant, DnaC$^{E170Q}$ (a Walker-B mutant), did not express using a periplasmic expression strategy. Expression of this construct was eventually worked out using the BL21AI expression strain (Invitrogen) grown in MDG-M9ZB medium (*Studier, 2005*). Freshly transformed cells were plated on 2xYT plates with 1% glucose to minimize leaky expression. Starter cultures were grown overnight in MDG minimal media and subsequently grown in M9ZB to OD ~2. Protein expression was induced with 0.2% arabinose overnight at 16°C. After expression, the material was processed as described for other DnaC mutants.

The purity of all DnaB and DnaC proteins throughout different purification steps was assessed by sodium dodecyl sulphate-polyacrylamide gel electrophoresis (SDS-PAGE). The concentration of the purified factors was determined using ND-1000 UV-Vis Spectrophotometer (Thermo Fisher) by measuring absorbance at 280 nm. Following purification, proteins were aliquoted, flash-frozen in liquid nitrogen, and stored at –80°C. Aliquots were thawed on ice for in vitro assays and used once; any remaining protein was discarded.

## ATPase activity assay

Measurement of DnaC ATPase activity was carried out as described previously (*Arias-Palomo et al., 2019*). Briefly, coupled ATP-hydrolysis assays were performed in 40 mM HEPES (pH 7.5), 10 mM MgCl$_2$, 5 mM dithiothreitol (DTT), and 0.1 mg/ml bovine serum albumin (BSA). A single Walker-A mutant of (K236A) proved to have a high residual ATPase activity background (not shown), so a double mutant was made in conjunction with catalytic glutamate substitution (E262A, referred to as 'DnaB$^{KAEA}$'). To assess DnaC activity, 5 µM of purified protein was mixed with 5.25 µM DnaB$^{KAEA}$ and 2.1 µg M13 ssDNA, 0.5 mM NADH, and 15 mM phosphoenolpyruvate in presence of a 4% v/v pyruvate kinase/lactate dehydrogenase enzyme mix (Sigma). Reactions were initiated by the addition of ATP to a final concentration of 25 µM to 6.4 mM, and the absorbance of the reaction was monitored at 340 nm in a 96-well, half-area plate (Corning) using a CLARIOstar (BMG Labtech) microplate reader at 37°C. An NADH standard curve was used to convert A$_{340}$ to NADH concentration, which in turn was used to calculate the rate of NADH loss that linearly correlates with ATP hydrolysis; initial rates were then calculated and plotted against ATP concentration. The data were fit to Michaelis-Menten equation using Prism to estimate k$_{cat}$ and K$_m$ values. The experimental data presented are from three independent experiments.

## DnaB loading assay

We utilized a gel-based assay with M13 ssDNA as a substrate to assess the efficiency of DnaB loading (*Fang et al., 1999*). Details of the assay have been described previously (*Arias-Palomo et al., 2019*). Briefly, purified DnaB$^{WT}$ or DnaB$^{KAEA}$ was mixed with equimolar amounts of wild-type or mutant DnaC to a final volume of 100 µl and a final concentration of 10 µM in presence of 40 mM HEPES (pH 7.5), 10 mM MgCl$_2$, 5 mM DTT, 0.1 mg/ml BSA, 5 mM ATP, and 1 µg M13 ssDNA. Reactions were incubated for 15 min at 30°C and then applied to a Bio-Gel A-1.5m (Bio-Rad) resin packed in a 0.7 × 15 cm Flex-Column (Kimble). The column was equilibrated and run in 40 mM HEPES-KOH (pH 7.5), 100 mM NaCl, 10 mM MgCl$_2$, and 5% glycerol; 250 µl fractions were collected by hand at 20°C, mixed with SDS-PAGE gel-loading dye, and resolved on 12% Tris-glycine SDS-PAGE gels. Gels were silver-stained, imaged on a Gel Doc EZ gel-documentation system (BioRad), and the amount of DnaB in each fraction estimated using Image Lab (BioRad). Helicase loading efficiency was calculated as the ratio of DnaB that eluted with ssDNA in the void volume of the column compared to the total amount of DnaB present in all fractions (*Figure 2—figure supplement 1B*). A few DnaC mutants (e. g., N203A or R220A) gave rise to a population of ssDNA-associated DnaB that migrated anomalously through the gel filtration matrix with a markedly delayed elution profile. These fractions were considered to contain aberrant or misloaded complexes and were not included in calculating the

amount of DnaB loaded compared to wild-type DnaC (which never displayed such behavior)—the DnaB present in these fractions was used for calculating the total amount DnaB only. The experimental data shown reflect the average of three independent experiments. Graphs were generated in Prism (GraphPad).

## DNA fork unwinding assay

Fork unwinding assay was performed as described previously (*Arias-Palomo et al., 2013*), with minor modifications. Briefly, high-performance liquid chromatography-purified DNA oligonucleotides were purchased from Integrated DNA Technologies. The forked DNA substrate was formed by annealing 20 µM 5'-Cy3-labeled oligonucleotide (5'-Cy3-TACGTAACGAGCCTGC(dT)25–3') with a 1.2 molar excess of a 3'-black hole quencher (BHQ)-labeled strand (5'-(dT)25-GCAGGCTCGTTACG TA-BHQ2-3') in annealing buffer (20 mM Tris-HCl (pH 7.5), 10 mM MgCl$_2$, and 20 mM KCl). A control 'captured' fork substrate with 100% maximum fluorescence signal was formed similarly to the forked DNA substrate, but that substituted the BHQ strand with an unlabeled capture strand (complementary to the base-paired region of the Cy3-labeled oligo, 5'-GCAGGCTCGTTACGTA-3'). The captured-fork control was used to normalize the signal between experiments. For the reactions, helicase and loader were premixed on ice, followed by the addition of substrate DNA. This reaction mixture was pre-warmed to 37°C; in parallel, the single-stranded capture oligo DNA was mixed with ATP and pre-warmed to 37°C separately. The ATP/capture-strand mixture was then added to the pre-warmed reaction mixture, pipetted gently to mix, and then three 20 µl aliquots were immediately pipetted into and read out from a 384-well black plate using a CLARIOstar (BMG Labtech) microplate reader at 37°C. The final reactions containing 200 nM DnaB hexamer (with or without 1.2 µM loader monomer), 100 nM fork substrate, 200 nM capture strand, and 1 mM ATP were monitored for fluorescence increase at 37°C for 26 min. The final assay buffer consisted of 20 mM HEPES-KOH (pH 7.5), 5 mM magnesium acetate, 50 mM potassium glutamate, 5% glycerol, and 0.2 mg/mL BSA. Reads from triplicate wells were averaged and analyzed in Excel (Microsoft) and normalized graphs were generated in Prism (GraphPad). Presented data are from three independent repeats.

## DnaC-ssDNA-binding assays

Fluorescence anisotropy was used to quantitate DNA-binding efficiency of DnaC variants. DnaC interaction assays with ssDNA were performed as previously described (*Arias-Palomo et al., 2019*; *Mott et al., 2008*). 0–50 µM DnaC was first mixed in a high-salt buffer (50 mM HEPES (pH 7.5), 500 mM KCl, 10% (v/v) glycerol, 10 mM MgCl$_2$) before being rapidly diluted fourfold into the binding buffer containing 50 mM HEPES (pH 7.5), 10% (v/v) glycerol, 5 mM MgCl$_2$, 2 mM ATP, 2 mM DTT, 0.1 mg/ml BSA, and 10 nM fluorescein-labeled dT25 oligonucleotide (IDT). Protein from the high-salt buffer was diluted in the binding reaction to bring the final salt concentration to 125 mM KCl. Aliquots of the samples were transferred to pre-warmed 384-well, black small-volume polystyrene plates (Greiner Bio-One). The plates were incubated for 10 min and then read in a CLARIOstar (BMG Labtech) microplate reader at 37° C. Averaged delta-anisotropy values were plotted against increasing DnaC concentrations. Data were fit to Hill equation for specific binding using Prism. The experimental data shown are from three independent repeats.

## PriC-dependent unwinding assays

Reactions were performed in triplicate using DNA substrates as previously described (*Wessel et al., 2013*) with the final monomeric concentrations of components as follows: 67 nM SSB, 120 nM DnaB, 400 nM DnaC (or variant), and 40 nM PriC. Briefly, a forked DNA substrate (60 bp duplex, 38 nt 3' and 5' overhangs) was made by annealing two oligonucleotides, one of which was labeled with [32]P. The substrate was preincubated with SSB (when included) for 3 min at 25°C. Reactions were initiated by the addition of DnaB, DnaC (or a variant), and PriC (when included) and incubated at 37°C for 30 min. Unwinding was terminated by the addition of 20 mM ethylenediaminetetraacetic acid, 0.5% SDS, 0.2 mg/ml of proteinase K, and 2.5 ng/ul of oligonucleotide 3 L-98 (final concentrations) and incubated at 37°C for 30 min. Samples were resolved on a 10% native polyacrylamide gel. The gel was then fixed in 10% methanol, 7% acetic acid, and 5% glycerol, dried, and exposed to a phosphorimager screen and imaged on Typhoon FLA 9000. Band intensities were quantified using

ImageQuant (GE Healthcare) and percent unwinding was determined by dividing the intensity of the single strand product band by the total intensity in the lane.

## Acknowledgements

We would like to thank current and former members of the Berger lab, in particular Ernesto Arias-Palomo and Frédéric Stanger, for helpful discussions during the analysis and preparation of data. This effort was supported by the NIGMS (R37-GM71747 to JMB and R01-GM098885 to JLK).

## Additional information

### Competing interests

Neha Puri: is affiliated with FogPharma. The author has no financial interests to declare. Valerie L O'Shea Murray: is affiliated with Saul Ewing Arnstein & Lehr, LLP. The author has no financial interests to declare. The other authors declare that no competing interests exist.

### Funding

| Funder | Grant reference number | Author |
| --- | --- | --- |
| National Institute of General Medical Sciences | R37-GM71747 | James M Berger |
| National Institute of General Medical Sciences | R01-GM098885 | James L Keck |

The funders had no role in study design, data collection and interpretation, or the decision to submit the work for publication.

### Author contributions

Neha Puri, Conceptualization, Formal analysis, Validation, Investigation, Visualization, Methodology, Writing - original draft, Writing - review and editing; Amy J Fernandez, Conceptualization, Formal analysis, Investigation, Methodology, Writing - original draft, Writing - review and editing; Valerie L O'Shea Murray, Conceptualization, Formal analysis, Investigation, Writing - original draft, Writing - review and editing; Sarah McMillan, Formal analysis, Investigation, Writing - original draft, Writing - review and editing; James L Keck, Formal analysis, Supervision, Funding acquisition, Visualization, Methodology, Writing - original draft, Writing - review and editing; James M Berger, Conceptualization, Resources, Data curation, Formal analysis, Supervision, Funding acquisition, Visualization, Methodology, Writing - original draft, Project administration, Writing - review and editing

### Author ORCIDs

Neha Puri  https://orcid.org/0000-0003-0079-7489
Amy J Fernandez  https://orcid.org/0000-0002-6130-3326
James M Berger  https://orcid.org/0000-0003-0666-1240

### Decision letter and Author response

Decision letter https://doi.org/10.7554/eLife.64232.sa1
Author response https://doi.org/10.7554/eLife.64232.sa2

## Additional files

### Supplementary files

• Transparent reporting form

### Data availability

All data generated during this study is included in the manuscript and supplementary files.

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
