## [Decision Letter]

Thank you for submitting your article "The molecular coupling between substrate recognition and ATP turnover in a AAA+ hexameric helicase loader" for consideration by *eLife*. Your article has been reviewed favorably by 3 peer reviewers, and the evaluation has been overseen by John Kuriyan as the Reviewing Editor and Senior Editor. The reviewers have opted to remain anonymous.

The reviewers have discussed the reviews with one another and the Reviewing Editor has drafted this decision to help you prepare a revised submission.

Summary:

This manuscript advances our understanding of loading factors for assembly of ring-shaped helicases around DNA. In particular, using *E. coli* DnaC as a model, the authors identify several key amino acids needed for ATPase function, and a new element for which they coin the term "arginine-coupler", that enables DnaC to couple productive ATPase action to the loading of the hexameric DnaB replicative helicase onto DNA. This resolved the long standing question of how DnaC retains bound ATP when complexed with DnaB helicase to form the full DnaC hexamer, yet only hydrolyzes the ATP when the DnaB6C6 complex binds to ssDNA. Interestingly, the report also identifies, for the first time, that the ATPase of the DnaB helicase participates in the loading reaction. Furthermore, the report also shows that DnaC can couple ATP to the unloading of the DnaB helicase; this is the first time that helicase unloading has been observed in a bacterial system.

The authors address DnaC function using assays for ATPase, helicase loading, helicase unwinding, helicase unloading, and an array of 5 mutants in DnaC's: Walker A, Walker B, Arg finger, Sensor I, and Sensor II motifs. The work is systematic, employs many biochemical assays, and joins the results with amino acid homologies and previously determined structures by this group and others. The AAA+ family of proteins is widely dispersed in nature, and they function in many different ways. Typically, AAA+ proteins function as oligomers, and DnaC is no exception. Hence, the findings of this report, while focused on the AAA+ DnaC protein, will have impact on other labs that work on AAA+ factors, and especially helicase and polymerase clamp loaders.

This report represents a significant advance in our knowledge of helicase loader function, and is quite suitable for publication in *eLife*, once the points made below are addressed in a revised manuscript.

Essential revisions:

The reviewers have identified the following major issues to address. For Point 2, if additional experimental results are available they should be included in the revised manuscript. If such data are not available, then the revised text should clearly point out the limitations of the analysis.

1. The primary discovery by Puri et al is the Arginine Coupler residue. However, the mechanism by which the Arg coupler controls ATPase activity in response to DNA binding is unclear from the discussion in the paper. Please provide a clearer and more comprehensive discussion of how how you envisage the coupling between DNA recognition and ATP hydrolysis to be accomplished.

2. It is unclear why a double mutant of DnaB was used to ablate ATPase activity. Mutation of the glutamate of the "Walker B" alone is sufficient to kill ATPase activity, as shown by other labs, but crucially will still allow ATP binding. As the authors themselves point out, mutation of the lysine of the "Walker A" may well affect ATP binding (K→R, although K→A is used here) as well as hydrolysis. The fact that significant ATPase activity persists in the R→A mutant suggests it does still bind at some level, and this seems an unnecessary complication. Although previous work is quoted to suggest ATP binding is not required for loading, it would be reassuring to see data showing ATP binding to DnaB (or hexamer formation?) has not been impaired since hexameric helicase rings are more more stable on nucleotide binding. This would make the results much cleaner. The use of just the E262A mutation would be even better and probably sufficient.

3. The work is interesting from a DnaB/C viewpoint but also has implications for related systems, as the authors point out. However, their comparisons with related systems show some relevant omissions whose inclusions would strengthen the comparisons. For example, the polymerase clamp loader RFC is compared but it has been shown previously that one of the conserved "RFC" motifs in an archaeal RFC plays a role in coupling ATP hydrolysis to PCNA clamp loading and when mutated the complex shows elevated ATPase but cannot release the clamp. There would seem to be some relevant comparisons with the roles of the mutations described here. Furthermore, mutation of the Arg finger in RFC creates a complex that binds ATP and PCNA but cannot load onto DNA substrates. This would seem to be complementary, and consistent, evidence to the role of the Arg coupler described in regulating a step in clamp opening.

4. The authors also make pertinent comments regarding the nature of the role of this AAA+ system (DnaC) as a one-off loading "switch" rather than a processive system (like a helicase). Similar arguments have been proposed for another "switch" in AAA+ proteins, namely the "Glutamate Switch" that prevents ATP hydrolysis until the assembly is complete and ready to carry out its reaction. There is (presumably) a Glutamate Switch in DnaC so what the interplay might be between these should be addressed in the discussion.

---

## [Author Response]

Essential revisions:The reviewers have identified the following major issues to address. For Point 2, if additional experimental results are available they should be included in the revised manuscript. If such data are not available, then the revised text should clearly point out the limitations of the analysis.1. The primary discovery by Puri et al is the Arginine Coupler residue. However, the mechanism by which the Arg coupler controls ATPase activity in response to DNA binding is unclear from the discussion in the paper. Please provide a clearer and more comprehensive discussion of how how you envisage the coupling between DNA recognition and ATP hydrolysis to be accomplished.

We regret this omission. We have added text to the Discussion that, together with the video, we believe clarifies this point (lines 400-403).

2. It is unclear why a double mutant of DnaB was used to ablate ATPase activity. Mutation of the glutamate of the "Walker B" alone is sufficient to kill ATPase activity, as shown by other labs, but crucially will still allow ATP binding. As the authors themselves point out, mutation of the lysine of the "Walker A" may well affect ATP binding (K→R, although K→A is used here) as well as hydrolysis. The fact that significant ATPase activity persists in the R→A mutant suggests it does still bind at some level, and this seems an unnecessary complication. Although previous work is quoted to suggest ATP binding is not required for loading, it would be reassuring to see data showing ATP binding to DnaB (or hexamer formation?) has not been impaired since hexameric helicase rings are more more stable on nucleotide binding. This would make the results much cleaner. The use of just the E262A mutation would be even better and probably sufficient.

We appreciate this question. A number of published studies that we drew on in planning for this work used Walker A substitutions when creating ATPase-defective mutants of *E. coli* DnaB. Fang et al. (Mol. Cell, 1999) reported that both K→R and K→A mutants are still capable of binding ATP with near wildtype affinity, and that these mutants retain the ability to be loaded onto *oriC* by DnaC. In our hands, the single mutants still displayed sufficient residual ATPase activity that they masked the ATPase activity of DnaC (consistent with their reported ability to bind nucleotide); thus, we added the Walker B E→A mutation to fully ablate hydrolysis by the helicase. The double DnaB mutant behaves identically to the wildtype protein during purification and migrates as a single hexamer by size exclusion chromatography in the absence of ATP, indicating that it is not destabilized by the substitution.

Nevertheless, we went ahead and generated and tested the single Walker B mutant of DnaB (E262A) in the loading assay. Overall, it behaved very similarly to the double mutant (see Author response image 1). As the outcome seems redundant to the findings obtained with the double mutant, we have elected not include this result in our studies; however, we can do so if the referees feel the distinction is important.

**Author response image 1. sa2fig1:** ssDNA loading assay with wildtype (WT) DnaC comparing WT DnaB vs. a single Walker B mutant (E262A) and a double Walker A/B mutant (KAEA) of the helicase.

3. The work is interesting from a DnaB/C viewpoint but also has implications for related systems, as the authors point out. However, their comparisons with related systems show some relevant omissions whose inclusions would strengthen the comparisons. For example, the polymerase clamp loader RFC is compared but it has been shown previously that one of the conserved "RFC" motifs in an archaeal RFC plays a role in coupling ATP hydrolysis to PCNA clamp loading and when mutated the complex shows elevated ATPase but cannot release the clamp. There would seem to be some relevant comparisons with the roles of the mutations described here. Furthermore, mutation of the Arg finger in RFC creates a complex that binds ATP and PCNA but cannot load onto DNA substrates. This would seem to be complementary, and consistent, evidence to the role of the Arg coupler described in regulating a step in clamp opening.

We agree that these are relevant comparisons and have noted them in the text (lines 316-318, RFC Arg-finger relationship; lines 486-490, archaeal RFC study).

4. The authors also make pertinent comments regarding the nature of the role of this AAA+ system (DnaC) as a one-off loading "switch" rather than a processive system (like a helicase). Similar arguments have been proposed for another "switch" in AAA+ proteins, namely the "Glutamate Switch" that prevents ATP hydrolysis until the assembly is complete and ready to carry out its reaction. There is (presumably) a Glutamate Switch in DnaC so what the interplay might be between these should be addressed in the discussion.

We appreciate the reviewers for catching this rather glaring omission. As it turns out it’s unclear from sequence analyses whether DnaC/DnaI protein universally retain a glutamate switch. Nonetheless, mention of this feature should have been made. A new supplemental figure (Figure S6) and accompanying text (lines 403-415) and figure legend have been added to the discussion.